# Gender and Tobacco Consumption among University Students

**DOI:** 10.3390/ijerph192214772

**Published:** 2022-11-10

**Authors:** Emília-Isabel Martins Teixeira-da-Costa, Maria-de-los-Angeles Merino-Godoy, Maria Manuela Monteiro Lopes Almeida, Alexandra Maria Monteiro Conceição Silva, Filipe Jorge Gamboa Martins Nave

**Affiliations:** 1Nursing Department, Health School, University of Algarve, 8000 Faro, Portugal; 2Health Sciences Research Unit: Nursing, 3000 Coimbra, Portugal; 3Nursing Department, Faculty of Nursing, University of Huelva, 21007 Huelva, Spain; 4Departamento de Saúde Pública e Planeamento, Administração Regional de Saúde do Algarve, IP, 8000 Faro, Portugal

**Keywords:** consumption of tobacco-derived products, gender, healthy settings, healthy universities, smoking cessation

## Abstract

In 2019, an estimated 155 million people aged between 15 and 24 were smokers. It is also known that 82.6% of current smokers started smoking between 14 and 25 years old. Tobacco uses in adolescents and young adults can lead to the development of serious and potentially life-threatening health problems. The aim of the present investigation is to identify and describe the practices related to the consumption of tobacco products and their distribution according to gender among students at the University of Algarve. This is an exploratory, cross-sectional study with a quantitative approach. For inferential statistics, a non-parametric analysis (χ^2^) was performed. The sample consisted of 326 university students, 75.5% female, with an average age of 26.03 years. In this sample, 45% of men and 57.7% of women reported never having smoked. In male students, the pattern of combined consumption is more frequent, with female students preferring conventional cigarettes. Statistically significant differences were found between genders for the pattern of tobacco consumption, the number of colleagues/peers who smoke, the opinion about tobacco-free outdoor spaces and the knowledge about new forms of tobacco/nicotine consumption. The university campus is identified by students as the second space where they most consume tobacco products and where they are most exposed to tobacco smoke. This fact forces a reflection on the strategies to be implemented to develop a healthier university.

## 1. Introduction

Smoking is one of the leading preventable causes of chronic illness, loss of quality of life and premature mortality [1]. It is estimated that in 2019, tobacco was responsible for 64.2% of deaths from cancer of the trachea, bronchi and lung; 48.5% of deaths from chronic obstructive pulmonary disease; 40.8% of deaths from esophageal cancer; 32.6% of deaths from cerebrovascular diseases; 13.1% of deaths from Alzheimer’s and other dementias; and 7.8% of deaths from diabetes [2]. Worldwide, tobacco use causes more than 7 million deaths per year [3]. If the pattern of smoking worldwide does not change, more than 8 million people a year will die from tobacco-related diseases by 2030, including an estimated 1.2 million deaths from exposure to second-hand smoke [4,5]. On average, a smoker dies 10 years earlier than a non-smoker [6].

According to estimates for 2019, smoking contributed to the deaths of more than 12,000 people residing in Portugal (10.43% of the total deaths). The age group from 50 to 69 was the most affected, with men showing the highest values [2]. According to the National Health Inquiry [7], in 2019, 16.8% of the population aged 15 and over were smokers (3 percentage points less than in 2014); 1.3 million people (14.2%) smoked daily and 248 thousand (2.8%) did so occasionally, 21.4% were former smokers and the majority (61.1%) had never smoked. Regular tobacco consumption recorded a ratio of 2 men for each woman.

It is worth noting that globally, in 2019, it was estimated that 155 million individuals aged between 15 and 24 were smokers, with a prevalence of 20.1% among men and 4.95% among women. The data show that 82.6% of current smokers started smoking between 14 and 25 years old and that 18.5% of smokers started smoking regularly at 15 years old [8].

Smoking consumption in adolescents and young adults is known to lead to the development of severe and potentially deadly health problems [9]. Nicotine exposure seems to have lasting effects on the development of the adolescent’s brain. Smoking also causes individuals in this age group to have less respiratory capacity and less resistance, which can affect athletic performance and other physical activities which are of great importance at this stage of life [10].

Entry into the university is, in fact, a moment of transition in the lives of individuals, with an important potential to lead to the adoption of new behaviors and lifestyles, to the reformulation of old behaviors or, eventually, to the validation of some existing ones. This marks a turning point in the biography of young people [11]. There is a need to develop or reinforce autonomy in many dimensions of life and expand decision-making processes regarding what they do with their time and with their body. These changes assume relevance to young people who are displaced from their usual residence. In this sense, higher education institutions are a space where students find new cultural and social contexts and where students are likely to reconfigure their social circles, symbolic pictures of reference and daily habits [11].

It should be noted here that the University of Algarve (UAlg) joined the Iberoamerican Network of Health Promoting Universities (RIUPS) in January 2017. The initiative of the Health Promoting Universities binds universities to generate an institutional political commitment to position Health Promotion as an integral element of the university’s vision, mission, values and strategic plan [12].

In this context, the area of smoking prevention has been developed for a few years within the university, through various awareness activities, developed in partnership with the Algarve Regional Health Administration and other partners, as is the case with the Company MADRE*fruta*©, which has been a key element in supporting the “cigarette fruit” initiative. This activity usually takes place in a common space on the university campus (a bar or lobby of a building) and consists of creating a moment of information and awareness, addressing the harmful effects of smoking and the process of smoking cessation. The members of the academic community that smoke are challenged to exchange their cigarettes for fruit; simultaneously, the evaluation of carbon monoxide in expired air is performed, and specialized information on this theme is given by health professionals of the regional tobacco prevention and control team. This kind of intervention is always enthusiastically received by the academic community. In the continuity of these activities, the possibility of creating a smoking cessation consultation available to the UAlg Academic Community was considered. However, before proceeding, we wanted to assess the smoking habits of this community; considering that there were no regional data on this population, we undertook this investigation. Being aware that health promotion interventions must respond to the needs of the target audience for which they are intended, we consider it relevant to assess this behavior from a gender-sensitive perspective. Our intention is reinforced by national data that indicate different patterns of consumption of tobacco/nicotine products between men and women and also by the health goal set by the General Health Directorate (DGS) to stop the increase in tobacco consumption among women [1].

## 2. Materials and Methods

This is an exploratory, cross-sectional study with a quantitative approach and a non-probabilistic sampling made for accessibility. The population is the Academic Community of the UAlg, constituted, to date, by approximately 8000 students. Data collection was carried out through the Google Docs platform, between 15 November 2019 and 2 February 2020. The form was sent via institutional email, with a reminder 30 days later. The sample (*n* = 326) had the following inclusion criteria: (a) being a student at the university; (b) voluntarily agreeing to participate in the study after observing the Free and Informed Consent Term (FICT). Participants with more than 20% missing data (10 participants) were excluded.

The aim of the present investigation is to identify and describe the practices related to the consumption of tobacco/nicotine products and their distribution according to gender among students at the UAlg. The drive is to recommend and implement, in the near future, gender-sensitive health promotion interventions in this area, namely, as previously mentioned, the availability of a smoking cessation consultation to the academic community. Targeting achieving the main objective of the study, based on the relevant theoretical framework [1], an ad hoc questionnaire was developed with several questions grouped into the following themes: sociodemographic characterization, consumption pattern, environmental exposure to tobacco, smoking policy, new forms of consumption and smoke cessation. In the elaboration of the data collection instrument, we had as a fundamental reference the National Program for the Prevention and Control of Tobacco, in its most recent editions—2017 and 2021 [1].

The statistical treatment of the data was performed with SPSS software (IBM® SPSS® Armonk, NY, USA) (Statistical Products and Service Solutions) v.28.0.0.0. First, a univariate analysis was performed, calculating descriptive statistics (standard deviation, mean and minimum and maximum values of quantitative variables). Regarding inferential statistics, we followed a non-parametric analysis (chi-square—χ^2^) given the nominal and ordinal nature of the variables under study, mainly through the analysis between genders that we proposed. The results of the chi-square test are presented in the tables only for the variables in which the differences found are statistically significant.

This investigation was approved on 6 November 2019 by the Data Protection Commission and the University of Algarve Rectory. All data obtained in this study were dealt with in accordance with Law No. 58/2019 of 8 August regarding the execution, in the national legal order, of the General Data Protection Regulation, and the study was carried out in accordance with the Declaration of Helsinki and Oviedo Convention, which establish international ethical criteria in the field of investigation with people.

This project was developed in partnership between the UALG+ healthy group and two members of the regional team of the National Smoking Prevention and Control Program, namely its coordinator and an environmental health member.

## 3. Results

### 3.1. Sociodemographic Characteristics

The sample consisted of 326 students from the University of Algarve, 246 women (75.5%) and 80 men (24.5%), with an average age of 26.03 years (25.82 women; 26.60 men); 88.3% were of Portuguese nationality, 81.9% were pursuing a bachelor’s degree or integrated master’s degree, 13.8% were post-graduate or master’s students, 2.5% were doctoral students and 1.8% were in other training paths (Table 1).

### 3.2. Tobacco/Nicotine Consumption

Analyzing tobacco/nicotine consumption habits, we find that 54.6% of students report never having smoked (in the group of men, 45% report never smoking; in the group of women, this value is 57.7%), 17.8% smoke occasionally, 14.7 % smoke daily and 12.9% consider themselves former smokers (Table 2).

Regarding the onset of tobacco consumption, 92.1% of those who smoke say that they started smoking at the age of 14 or more years, and 7.9% state that they started this type of consumption between 12 and 13 years of age. Performing the analysis according to gender, we find that 3.8% of men and 3.3% of women refer to the first time being between 12 and 13 years of age, and 47.5% of male students and 36.6% of female students report having started smoking habits at the age of 14 or more years.

Regarding the usual consumptions, as can be appreciated in Table 3, we find that the most consumed tobacco type, for the total sample, is conventional cigarettes (manufactured or hand-rolled) with values of 18.4% and 9.8% of questioned students having a type of combined consumption, using various types of tobacco/nicotine products. Examining the different genders separately, we find that for men, the largest percentage (18.8%) falls on the combined consumption of various tobacco/nicotine products and the largest percentage of women (18.7%) is for smoking conventional cigarettes.

For this variable (tobacco consumption pattern) after the application of the chi-square test, there was a statistically significant difference in relation to gender, with *p* = 0.004 and χ^2^ = 17.122.

When we asked about the frequency of consumption of tobacco products in the last 30 days, we found that 61.2% of respondents (smokers) reported having smoked 10 or more times, 11.2% smoked 3 to 9 times and 27.6% smoked 1 or 2 times. It was also found that of the surveyed students (smokers) who reported smoking every day, 52% smoked 10 or more cigarettes a day.

Some of the respondents (20.9%) identify themselves as occasional smokers, of which 29.4% smoke more than once a week but not every day, 26.5% report smoking once a month, 16.2% report smoking less than once a week, 5.9% smoked once a week and 22.1% reported another unspecified situation. Observing the genders separately regarding occasional consumption of tobacco/nicotine products, we found that, in men, the most frequent situation is once a month with 8.8%, and for women, the highest percentage is for smoking more than once a week, but not every day (5.7%).

### 3.3. Second-Hand Smoke

Identifying the place where tobacco/nicotine products are usually consumed also allows the characterization of their consumers. In this way, and given that several answers were possible, we could see, as can be perceived in Figure 1, that the places where tobacco consumption is more notorious are social events (parties, bars, clubs) in the first place, with 82.9% of responses, university outdoor spaces in second place with 57.1% and home in third place with 43.8%.

Regarding environmental exposure to smoke from tobacco/nicotine products, we found that 40.8% of respondents indicate being exposed almost every day, 36.8% report being exposed sometimes, 20.9% report being exposed rarely and only 1.5% of respondents mention never being exposed. For this question, the percentage differences between the two genders are minimal.

In addition, when we identified the spaces where environmental exposure takes place (also here multiple answers were possible) we notice that the most mentioned places are social events (parties, bars, clubs) with 83.7% of responses, the university (open spaces) with 62.3% and public spaces such as parks with 59.5%. Only 7.1% of respondents say they are not exposed to tobacco product smoke (Figure 2).

We, similarly, wanted to know how often students saw professors, staff or other students smoking at the entrance to the buildings in the spaces of the university campus. We similarly verified here that the percentage differences between the two genders were minimal, and, as can be seen in Figure 3, the group that is most often seen smoking at the entrance of buildings is students.

It was also our goal to evaluate the tobacco/nicotine product consumption behavior in the group of pairs of students surveyed by asking them what percentage of their friends smoked. We find that 8.9% of students report that almost all their friends smoke, 41.1% report that many of their friends smoke, 40.8% report that few of their friends smoke and only 9.2% of students say that no one in their group of friends smokes. In the group of female students, 9.8% state that none of their friends smoke, and this value is 7.5% in the group of male students; 45.5% of women and 26.3% of men indicate that few of their friends smoke, 35.4% of women and 58.8% of men indicate that many of their friends smoke and 7.5% of men and 9.3% of women indicate that almost all their friends smoke (Table 4).

After applying the chi-square test for this variable (number of friends/colleagues who smoke), we found that there are statistically significant differences regarding gender, with *p* = 0.003 and χ^2^ = 14.096.

### 3.4. Smoke Policy

Considering smoke policies, we intended to determine if students agree with the existence of outdoor spaces without smoking; we found (Table 5) that 78.8% of men and 91.9% of women agree with the existence of this type of space, and the value for the total group of students was 88.7%.

For this variable (opinion about smoke-free outdoor spaces), after application of the chi-square test, we verified there are significant differences in relation to gender, with *p* = 0.001 and χ^2^ = 10.328.

We also wanted to understand the position of respondents regarding the creation of outdoor spaces intended for smokers, and we observed that 83.8% of men and 88.2% of women agree, both values being very close to the value of 87.1% that arises for the total group of respondents.

### 3.5. New Forms of Consumption

We wanted to appreciate, as well, the level of knowledge of respondents about new forms of tobacco/nicotine consumption. We could attest that 11.3% of men and 3.7% of women consider that new consumption devices are less harmful than conventional cigarettes, 13.8% of men and 25.2% of women believe that these new devices are as harmful as conventional cigarettes, 2.5% of men and 4.5% of women indicate that these new formats are more harmful than the classic modality of consuming tobacco and 72.5% of men and 66.7% of women believe that these new forms lack research that allows for more robust conclusions to be drawn (Table 6).

For this variable (new forms of tobacco/nicotine consumption) after application of the chi-square test, we verified there are statistically significant differences in relation to gender, with *p* = 0.013 and χ^2^ = 10.727.

### 3.6. Smoking Cessation

When students were asked about what would motivate them to stop smoking, the values shown in Figure 4 were obtained; for the group of smokers, the highest percentages are obtained with individual health as motivation, with 27.2% for women and 33.8% in men. This was followed by public health as motivation, with 7.3% for women and 12.5% for men. It also seems that rising tobacco prices can also have a motivating effect for quitting smoking for 3.3% of women and 5% of men.

We also wanted to know the percentage of smoking respondents who would be willing to stop smoking; we obtained a positive answer from 54.5% of men and 50% of women. Regarding whether expert help for smoking cessation would be accepted, we found that 46% of men who smoke and 42.9% of women who smoke indicate that they would be receptive to professional monitoring in the smoking cessation process.

## 4. Discussion

According to the 2020 Report of the National Program for the Prevention and Control of Tobacco Use, there has been a decrease in tobacco consumption in Portugal in the last five years; however, the General Directorate of Health continues to emphasize the importance of smoking prevention in the younger age groups, an intention that is clear in its strategic guidelines for the next two years [13].

The aim of this project is to identify and describe tobacco consumption practices among university students and particularly to recognize possible nuances in smoking habits according to gender. Although some of the variables analyzed did not show statistically significant differences between genders, we believe that the data deserve to be carefully observed to allow drawing up lines of intervention that are better directed to the real needs of the community for which they are intended. Most of the sample consisted of women, approximately 26 years old, Portuguese and pursuing a bachelor’s degree or integrated master’s degree.

The fact that we have a much more significant percentage of female than male participants reflects what is happening in the academic community of UAlg, which has a higher number of female students than male students, with an approximate gender difference of 18% [14]. We can justify a more marked gender difference in our study than in the UALg population based on the data collection method, as the evidence seems to indicate that women are more willing to answer online questionnaires than men [15].

Regarding consumption, we found that the average values found are higher than those recorded for the general population in the last Report of the National Program for the Prevention and Control of Tobacco 2020 [13], translating into 32.5% of smokers in our sample compared to 16.8% in the general population. However, according to the same source, when we look at the age group to which the average age of our sample corresponds, we find a percentage value closer to that found by us (27.6% in the age group from 25 to 34 years old). It should also be noted that the Algarve region is the third region in the country with the highest prevalence of smokers, another fact that may contribute to our findings [13]. Another study carried out in Portugal with the university population [11] also found higher values than those presented in the aforementioned report, and closer to ours, in the order of 26.5%. It should also be noted that these authors [11] show that first-year university students are those with the highest values (31.7%) of consumption of tobacco products, and even though we did not inquire about the year of attendance in our study, by taking into account that the modal value for age is 19, we can estimate that we have in the sample an important number of students in the initial years of their degrees.

There is a consensus among the consulted authors [16,17,18] regarding the beginning of smoking, pointing to adolescence and even mentioning that few people become smokers after the age of 18, also underlining the important role of the peer group in early experimentation. The data we found in our sample show that the vast majority (92.1%) of the students indicate having smoked for the first time when they were 14 or over and only 7.9% started this consumption between 12 and 13 years old, with no statistically significant differences regarding gender. These data are close to those found for the general population [13].

Although the general consumption of tobacco has declined in recent years, there is an increase in the consumption of new tobacco products [13]. We found in our study that students continue to consume mostly conventional cigarettes but are also users of a combined consumption of various tobacco/nicotine products. For this variable, we found statistically significant differences between the genders, verifying that the combined consumption of tobacco/nicotine products is more evident in male students (18.8%) and female students preferentially opt for conventional cigarettes (18.7%). This consumption pattern is also consistent with data found at national [13] and international [19] levels. It should be noted here that the evidence points to the fact that young people who use various tobacco/nicotine products are at greater risk of developing nicotine dependence and may be more likely to continue to use tobacco into adulthood [10,18].

In the sample studied, we found that 45.2% of smokers reported smoking daily (14.7% of the sample) and 54.8% identified themselves as occasional smokers (17.8% of the sample). No significant differences were found between genders. The values found in our study for individuals who report smoking on a daily basis are close to those found by the General Directorate of Health (DGS) [13], and those regarding occasional smokers are slightly higher than what we find in the literature [1,13]. Two main aspects can justify these findings: the study was carried out in one of the regions of the country with the highest prevalence of smoking, and the vast majority of respondents were in the first years of their degrees—a fact that, as we have already mentioned, may indicate a higher level of consumption of tobacco/nicotine products [11].

Exposure to environmental tobacco smoke poses serious health risks for passive smokers. It is especially associated with the development of several health problems, especially cardiovascular pathologies [20]. In the present study, 40.8% of respondents report that they are exposed to tobacco smoke practically every day (with no statistically significant differences between genders). These values are much higher than those presented in the DGS report for the general population (7.7%) [13]; they are similar to the values indicated by this entity in relation to leisure facilities, and even so, in our sample, they are higher.

Although smoking is prohibited inside the university buildings, individuals who smoke often choose to do so near the entrance doors, and these areas are often covered, which potentially increases the concentration of harmful substances in the air breathed in these places. Passive smoking in covered outdoor areas, even in the open, contributes to the increase in toxic particles in the air we breathe, often exceeding the 24 h reference value established by the World Health Organization. These conditions are especially worse in rainy or cold weather, as smokers are concentrated under the cover and therefore closer to the entrance. This smoke-laden air can be transferred from the entrance to the internal area of the building, reducing the quality of the air breathed in these spaces as well [20].

By analyzing the data obtained, we could see that the university is a space of risk in terms of consumption and exposure to tobacco smoke. This fact concerns us and justifies the relevance of analyzing this phenomenon in this context and defining an urgent intervention plan. The outdoor spaces (in the open) of the university campus are identified by smoking students as the second place where they most consume tobacco/nicotine products (57.1%), they are also pointed out by all respondents as the second place where they are most exposed to tobacco smoke (62.3%). It is considered that there are no safe levels of exposure to second-hand smoke, and the recommendation is that it should be reduced to the minimum level possible [21]. In this context, it is urgent to find strategies that allow reducing the spaces of exposure to tobacco smoke within the university campus.

In this context, we also found that students are receptive to the creation of smoke-free outdoor spaces on campus, given that 78.8% of men and 91.9% of women are of the opinion in this regard. Here we found statistically significant differences regarding gender, noting that both are in favor of this change, but female students are the ones who express a stronger opinion in this regard. This fact may be related to a greater proactivity in the search for health-promoting circumstances, manifest in the data that attest that women tend to have higher rates of use of health services, in general, and specifically a greater use of smoking cessation services, according to data found in several countries [22,23,24].

Considering the students’ opinions about the new forms of tobacco/nicotine consumption, statistically significant differences were found according to gender, with women being the ones who most strongly say that new forms of tobacco/nicotine consumption are as harmful as conventional tobacco. This aspect may be related to the levels of health literacy, with some authors documenting higher levels of health literacy in women, namely in the dimension of health knowledge [25,26].

We found that more than half of both men and women with smoking habits would be willing to give up smoking. Similar data were found in other studies [27,28,29] and should be seen as an opportunity for the development of tobacco prevention and control interventions among young people. In addition to these data, regarding the receptivity to giving up smoking with specialized help, 46% of male students and 42.9% of female students demonstrate that they would be receptive to professional assistance in this process. This is another intervention juncture that cannot be neglected.

Regarding the sources of motivation for smoking cessation, we found that individual health and collective health seem to be the main reasons for quitting smoking, followed by the increase in tobacco prices; although we did not find significant differences between genders here, males presented higher percentage values in these three points. Particularly, individual health appears consistently, throughout several studies, as a motivation to quit smoking [30,31].

It should be noted that the evidence found in several countries shows that increasing the price of cigarettes is effective in reducing the prevalence of tobacco consumption in young people and adults. Higher prices of tobacco products encourage cessation and prevent initiation of tobacco use, with younger age groups being more sensitive to price increases than adults [22,32].

## 5. Conclusions

The pattern of tobacco consumption found in the present investigation points to values that are higher when compared to national data for the general population but are similar to those found by other authors when framed in the age group, academic profile and region of the country. We did not find statistically significant differences between genders regarding the frequency of smoking, noting that 45.4% of the students surveyed currently smoke or have smoked at some point in their lives. There were also no differences between genders regarding environmental exposure to tobacco product smoke, intention to quit smoking and sources of motivation for smoking cessation. Statistically significant differences were found between genders for the pattern of tobacco consumption (products consumed), the number of colleagues/peers who smoke, the opinion about tobacco-free outdoor spaces and the knowledge about new forms of tobacco/nicotine consumption.

It should be noted that the university campus is identified, in this study, by students as the second space where they most consume tobacco products and where they are most exposed to tobacco smoke. This fact forces a reflection on the strategies to be implemented to develop university spaces where it is easier to be healthy.

The data presented here must be interpreted considering their limitations. One of the limitations is that smoking status was self-reported, and therefore the data are subject to response bias. In addition, a larger sample size with a gender distribution closer to the characteristics of the academic population will be desirable in future studies. The present study will serve as a preliminary finding, and we expect to apply it again periodically when interventions to promote a progressively smoke-free university are implemented.

## Figures and Tables

**Figure 1 ijerph-19-14772-f001:**
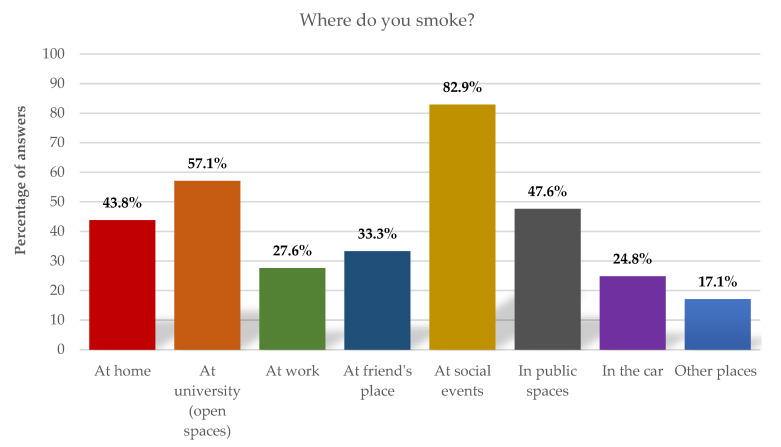
Tobacco consumption places.

**Figure 2 ijerph-19-14772-f002:**
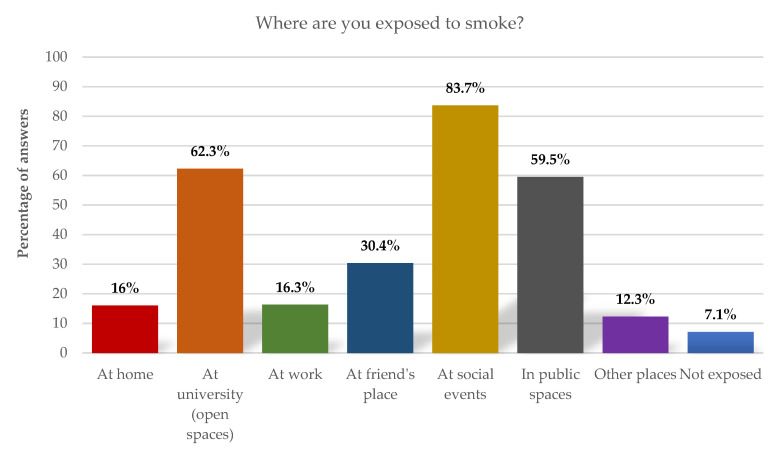
Places of exposure to tobacco smoke.

**Figure 3 ijerph-19-14772-f003:**
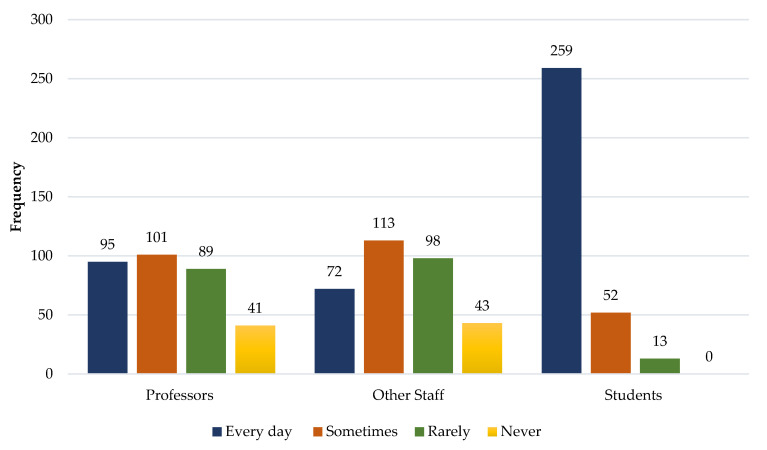
Frequency at which professors, other staff and students smoke at the entrance doors of university buildings.

**Figure 4 ijerph-19-14772-f004:**
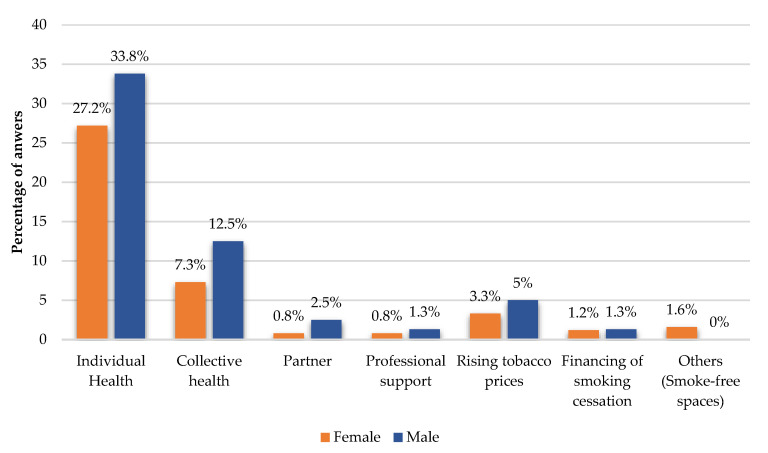
Sources of motivation to quit smoking.

**Table 1 ijerph-19-14772-t001:** Sociodemographic characteristics of students.

	*n* = 326 *n* (%) Average (SD)
**Gender**	
Male	80 (24.5%)
Female	246 (75.5%)
**Age**	26.03 (8.198%)
Range 18–60 years	
**Nationality**	
Portuguese students	288 (88.3%)
Foreign students	38 (11.7%)
**Degree**	
Bachelor’s degree or integrated master’s degree	267 (81.9%)
Post-graduate or master’s students	45 (13.8%)
Doctoral students	8 (2.5%)
Other	6 (1.8%)

Abbreviation: SD, standard deviation.

**Table 2 ijerph-19-14772-t002:** Smoking habits.

Habits Gender % (*n*)		I Smoke Every Day	I Smoke Occasionally	I Don’t Smoke Anymore	I Never Smoked	Total
Within group	Male	18.8% (15)	25.0% (20)	11.3% (9)	45.0% (36)	100% (80)
	Female	13.4% (33)	15.4% (38)	13.4% (33)	57.7% (142)	100% (246)
Within Total		14.7% (48)	17.8% (58)	12.9% (42)	54.6% (178)	100% (326)

**Table 3 ijerph-19-14772-t003:** Tobacco consumption pattern.

Consumption Pattern Gender % (*n*)		Non-Smoking	Cigarettes (Manufactured or Hand Rolled)	Other Types of Tobacco (Cigarillos, Pipes, Cigars)	Heated/ Non-Burning Tobacco (e.g., IQOS)	Electronic Cigarettes	Other Types of Combined Consumption	Total
Within group	Male	56.3% (45)	17.5% (14)	2.5% (2)	1.3% (1)	3.8% (3)	18.8% (15)	100% (80)
	Female	71.1% (175)	18.7% (46)	0.4% (1)	2.0% (5)	0.8% (2)	6.9% (17)	100% (246)
Within Total		67.5% (220)	18.4% (60)	0.9% (3)	1.8% (6)	1.5% (5)	9.8% (32)	100% (326)
Chi-square		χ^2^ = 17.122	*p* = 0.004					

**Table 4 ijerph-19-14772-t004:** Number of friends/colleagues who smoke.

Number of Friends Gender % (*n*)		Nobody	Few (<10%)	Many (11–75%)	Almost Everyone (>75%)	Total
Within Group	Male	7.5% (6)	26.3% (21)	58.8% (47)	7.5% (6)	100% (80)
	Female	9.8% (24)	45.5% (112)	35.4% (87)	9.3% (23)	100% (246)
Within Total		9.2% (30)	40.8% (133)	41.1% (134)	8.9% (29)	100% (326)
Chi-square		χ^2^ = 14.096	*p* = 0.003			

**Table 5 ijerph-19-14772-t005:** Opinion about smoke-free outdoor spaces.

Gender % (*n*)		Do You Agree with the Existence of Smoke-Free Outdoor Spaces?		Total
		Yes	No	
Within Groups	Male	78.8% (63)	21.3% (17)	100% (80)
	Female	91.9% (226)	8.1% (20)	100% (246)
Within Total		88.7% (289)	11.3% (37)	100% (326)
Chi-square		χ^2^ = 10.328	*p* = 0.001	

**Table 6 ijerph-19-14772-t006:** New forms of tobacco/nicotine consumption.

Gender % (*n*)		Do You Consider That New Consumption Devices (Electronic Cigarettes and Heated/Non-Burning Tobacco)				Total
		Are Less Harmful Than Conventional Cigarettes	Are as Harmful as Conventional Cigarettes	Are More Harmful Than Conventional Cigarettes	Need More Research	
Within Groups	Male	11.3% (9)	13.8% (11)	2.5% (2)	72.5% (58)	100% (80)
	Female	3.7% (9)	25.2% (62)	4.5% (11)	66.7% (164)	100% (246)
Within Total		5.5% (18)	22.4% (73)	4.0% (13)	68.1% (222)	100% (326)
Chi-square		χ^2^ = 17.122.	*p* = 0.004			

## Data Availability

The data presented in this study are available on request from the corresponding author. The data collection tool is also available on request from the corresponding author. The data are not publicly available due to privacy restrictions.

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
