# Peer review of "Gender and Tobacco Consumption among University Students"

_ijerph, 2022, doi:10.3390/ijerph192214772_

Round 1
Reviewer 1 Report
General comments:
The topic and purpose of the study are relevant and current.
In the abstract, the aim of the study should be defined as presented in section 2 (material and methods).
It may be pertinent to present the results regarding the sociodemographic characteristics of the population in a table, as was done for the other results.
Specific comments:
line 423-424 - Failure to obtain access to information
row 437-438 - Failure to obtain access to information
Reviewer 2 Report
Please see detailed comments on the attached document.

Round 2
Reviewer 2 Report
The paper is much improved. A final minor English/spell check is required.